# Fabrication of Orange Fluorescent Boron-Doped Graphene Quantum Dots for Al^3+^ Ion Detection

**DOI:** 10.3390/molecules27196771

**Published:** 2022-10-10

**Authors:** Weitao Li, Luoman Zhang, Ningjia Jiang, Yongqian Chen, Jie Gao, Jihang Zhang, Baoshuo Yang, Jialin Liu

**Affiliations:** 1Textile and Garment Industry of Research Institute, Zhongyuan University of Technology, Zhengzhou 450007, China; 2Institute of Nanochemistry and Nanobiology, School of Environmental and Chemical Engineering, Shanghai University, Shanghai 200444, China

**Keywords:** graphene quantum dots, fluorescence quenching, ion detection, p-phenylenediamine, Aluminum ion detection

## Abstract

Aluminum is a kind of metal that we often encounter. It can also be absorbed by the human body invisibly and will affect our bodies to a certain extent, e.g., by causing symptoms associated with Alzheimer’s disease. Therefore, the detection of aluminum is particularly important. The methods to detect metal ions include precipitation methods and electrochemical methods, which are cumbersome and costly. Fluorescence detection is a fast and sensitive method with a low cost and non-toxicity. Traditional fluorescent nanomaterials have a high cost, high toxicity, and cause harm to the human body. Graphene quantum dots are a new type of fluorescent nanomaterials with a low cost and non-toxicity that can compensate for the defects of traditional fluorescent nanomaterials. In this paper, c-GQDs and o-GQDs with good performance were prepared by a bottom-up hydrothermal method using o-phenylenediamine as a precursor and citric acid or boric acid as modulators. They have very good optical properties: o-GQDs exhibit orange fluorescence under UV irradiation, while c-GQDs exhibits cyan fluorescence. Then, different metal ions were used for ion detection, and it was found that Al^3+^ had a good quenching effect on the fluorescence of the o-GQDs. The reason for this phenomenon may be related to the strong binding of Al^3+^ ions to the N and O functional groups of the o-GQDs and the rapid chelation kinetics. During the chelation process, the separation of o-GQDs’ photoexcited electron hole pairs leads to their rapid electron transfer to Al^3+^, in turn leading to the occurrence of a fluorescence-quenching phenomenon. In addition, there was a good linear relationship between the concentration of the Al^3+^ ions and the fluorescence intensity, and the correlation coefficient of the linear regression equation was 0.9937. This illustrates the potential for the wide application of GQDs in sensing systems, while also demonstrating that Al^3+^ sensors can be used to detect Al^3+^ ions.

## 1. Introduction

Aluminum is the most abundant metal element in the earth’s crust. It has many excellent characteristics, such as high strength, good electrical conductivity, good thermal conductivity, and good ductility [1]. Therefore, it can be widely used in various fields and applications, such as in aircrafts, automobile manufacturing, and aluminum foil packaging. However, aluminum also poses risks to the human body; for example, the widespread use of aluminum-purified water causes brain damage and memory loss, which is also an early feature of Alzheimer’s disease. Therefore, it is very important to detect aluminum in order to reduce its intake into the human body. The methods for detecting metal ions include electrochemical methods, precipitation methods, spectral methods, chromatography, and so on [2,3]. These methods can effectively detect metal ions, but their operation is complex and costly, involves expensive equipment, and they cannot quickly and sensitively identify metal ions. A fluorescent probe is a simple, highly specific, and highly sensitive material that can quickly detect metal ions, and its low cost and high selectivity support its wide use in ion detection [4,5,6].

Graphene quantum dots represent a new type of fluorescent nanomaterial with excellent optical properties and low toxicity, which has successfully become an ideal substitute for traditional fluorescent materials [7,8]. Graphene quantum dots possess tunable optical properties due to their unique electronic structure, stable and durable fluorescence, controllable emission wavelengths, and wide excitation wavelength range, which have been applied in photocatalysis, sensors, solar light sources, organic LEDs, photodetectors, and in other fields/applications [9,10,11]. In addition, graphene quantum dots have simple functions; excellent thermal, mechanical, and electrochemical luminescence properties; low toxicity and high environmental safety; and good biocompatibility, and have been successfully applied in ion detection, organic detection, biological imaging, drug delivery, and other fields [12,13,14].

Graphene quantum dots can be prepared by a variety of methods, including top-down and bottom-up methods. The top-down methods include chemical oxidation, acid oxidation, etc., while the bottom-up methods include hydrothermal methods, electrochemical methods, and so on [15,16]. The top-down approach uses physical and chemical methods to cut large precursors such as graphene carbon into nanoscale GQDs, while the bottom-up approach uses small molecules as precursors and synthesizes GQDs through chemical reactions [17,18]. The bottom-up approach is more controlled than the top-down process, using small molecules as precursors to synthesize GQDs through a number of chemical reactions. The reaction process is more controllable, and the prepared GQDs have better performance and a higher yield [17,18,19,20]. This study adopts the bottom-up method of hydrothermal method, since its operation process is simple, it causes low levels of pollution in the environment, it is cheap, and it can efficiently prepare GQDs.

Due to their electrochemical characteristics, GQDs can are very suitable for application in the field of sensing, where it can be used as a fluorescent probe for the selective detection of metal ions [21,22]. In addition, GQDs have good water solubility, low toxicity, will not change the water quality during metal detection, are harmless to the human body, and their use in the synthesis of fluorescent probes is simple, but they also have a strong optical stability [23]. When a fluorescent probe is used for metal ion detection, the metal ions combine with the functional ligand on the surface of the GQDs to change the fluorescence intensity of the GQDs, leading to fluorescence quenching and achieving the metal ions’ detection [24,25]. Our research group previously reported full-spectrum fluorescent GQDs. In the experiment, the particle size and fluorescence wavelength of the GQDs were successfully adjusted by the bottom-up hydrothermal method with o-phenylenediamine(o-PD) as the precursor and acid as the regulator, and panchromatic fluorescent GQDs were prepared and successfully applied to LEDs. Inspired by this, boric acid and citric acid were added as acid modulators to regulate the performance of GQDs, which enhanced the fluorescence properties and various properties of the GQDs. Among them, the GQDs prepared by adding boric acid can selectively recognize Al^3+^ ions, which can be successfully applied to the detection of Al^3+^. P-phenylenediamine (p-PD) is an organic compound and one of the simplest aromatic diamines; it is also a widely used intermediate, can be used in polymers, rubber, aging agents, photo developers, and so on; p-phenylenediamine can also be used as a sensitive reagent to detect iron and copper. Boric acid (BA) is an inorganic compound used as a preservative disinfectant. The GQDs synthesized by p-phenylenediamine and boric acid have good ligands and can be detected by combining with metal ions.

In this experiment, p-phenylenediamine and boric acid or citric acid were used as precursors, and two GQDs were synthesized by the bottom-up hydrothermal method, named o-GQDs (BA was used as the regulator) and c-GQDs (CA was used as the regulator), respectively. The prepared c-GQDs emitted green light under the irradiation of a UV lamp, while the o-GQDs emitted bright orange fluorescence. They had excellent fluorescence performance, good dispersion, and a uniform size distribution. In particular, the o-GQDs were successfully doped with boron to enhance their various properties, and this technique was successfully applied to the detection of Al^3+^ ions in the subsequent detection of metal ions. The fluorescence quenching of o-GQDs by Al^3+^ is due to the obvious effect, which may be related to the strong binding and fast chelation kinetics between the N and O functional groups and the ions of o-GQDs. During the chelation process, the separation of the o-GQDs’ photoexcited electron hole pairs leads to rapid electron transfer, which promotes the electron/hole recombination process, leading to the arrest of electron transition, and finally leading to the quenching of the o-GQDs’ fluorescence signal [26,27]. The experimental process is shown in Figure 1.

## 2. Results and Discussion

### 2.1. Morphology Characterization of GQDs

Figure 2 shows the morphologies of the c-GQDs and o-GQDs. During the test, about 500 samples were selected to draw the height distribution histogram of the AFM and the relative size distribution histogram of the TEM. Figure 2a,b shows the AFM diagram and thickness distribution diagram of the c-GQDs and o-GQDs. It can be seen from the figure that the average thickness of the c-GQDs and o-GQDs is 3.53 nm and 1.23 nm, indicating that they are composed of several layers of graphene. Figure 2c,d shows the TEM and size distribution of the c-GQDs and o-GQDs. As can be seen from the figure, the two GQDS are evenly dispersed. The average particle size of the c-GQDs is 0.2 nm, and that of the o-GDs was 0.23 nm. The o-GQDs are larger than the c-GQDs, indicating that the regulation of boric acid is beneficial to the growth of GQDs. Figure 2e,f shows high-resolution TEM images of the c-GQDs and o-GQDs. The illustration shows Fourier transform images. The crystal morphology of the c-GQDs and o-GQDs can be clearly seen from the high-resolution TEM images, and a single crystal structure can be found. The lattice spacing of the two GQDs is 0.23 nm. It can be seen from the Fourier transform figure that the regular hexagonal symmetry points in the figure are similar to the benzene ring lattice structure of graphene, indicating that c-GQDs and o-GQDs have good single crystal structures. By contrast, the o-GQDs have better lattice symmetry, indicating that they have better crystallinity, which may be the reason for their better optical performance.

### 2.2. Structural Characterization of GQDs

The Figure 3a shows the XRD pattern of the o-GQDs. The o-GQDs have an obvious diffraction peak with a layer spacing of 3.11 Å at about 29.6°. In addition, the c-GQDs have an obvious wide diffraction peak near 29.1°, and its layer spacing is 3.05 Å (Appendix A). The surface spacing of o-GQDs is larger, which may be due to the B-doping. Figure 3b shows the FT-IR diagram of the o-GQDs; there is a strong stretching vibration peak O-H bond at 3340 cm^−1^ [28,29]. Before the infrared test, the sample was completely dried, indicating that the O-H comes from the o-GQDs. There is a strong stretching vibration peak at the N-H bond at 3211 cm^−1^, and the bond energies at 1613 cm^−1^, 1515 cm^−1^, and 1303 cm^−1^ represent C=C, B-O, and B-C, respectively [30,31,32]. These functional groups indicate that there are many amino and hydroxyl groups on the surface of the o-GQDs, so the o-GQDs have a good hydrophilic property. In addition, B-O functional groups and B-C functional groups were detected, indicating that the boron was well doped into the o-GQDs. Fourier transform infrared spectroscopy of the c-GQDs shows that they have similar groups to o-GQDs (Appendix A). Figure 3d shows the Raman diagram of o-GQDS. The D-peak of the disordered sp^3^ hybrid carbon structure (1355 cm^−1^) and the G-peak of the ordered sp^2^ graphite carbon structure (1523 cm^−1^) can be clearly observed. The D-peak (1347 cm^−1^) and G-peak (1550 cm^−1^) of the c-GQDs can be observed from Appendix A
Appendix A. The I_D_/I_G_ ratio of the o-GQDs is higher than that of the c-GQDs. Figure 3c is a color image of the ratio of the G-peak to the D-peak of o-GQDS. Appendix A is a color diagram of the ratio of the G-peak to the D-peak of the c-GQDs. Figure 3d and Appendix A are the Raman spectra of the box area of their color maps. In addition, the I_D_/I_G_ of the o-GQDs and c-GQDs are 0.894 and 0.879, respectively. The I_D_/I_G_ of the o-GQDs is slightly higher than that of the c-GQDs, which may be due to the sp^3^ hybridization of the B-doped o-GQDs, which makes the peak D value higher; consequently, the graphitization degree of the o-GQDs is not as good as that of the c-GQDs.

To determine the surface composition of the GQDs, XPS spectra were further tested. From the XPS total spectrum in Figure 4a, the o-GQDs have four peaks: C1s (284 eV), N1s (398 eV), O1s (532 eV), and B1s (193 eV). This indicates that B exists in the graphene quantum dots after the addition of boric acid. This is consistent with the FT-IR data. Figure 4b–e is the fine spectrum of C1s, N1s, O1s, and B1s of the fitted o-GQDs. Its fine spectrum of C1s can be divided into five peaks, corresponding to C-B (284.19 eV), C-H (284.6 eV), C-N (285.07 eV), C-O/C=O (285.81 eV), and C-O (286.9 eV) [33,34,35]. The fine spectrum of N1s has three peaks, which are N-B (398.08 eV), N-H (399.53 eV), and N-C (400.35 eV) [35,36,37]. The fine spectrum of O1s has four peaks, namely, C-O-C (530.22 eV), O-H (531.16 eV), C-O-O/C-OH (532.26 eV), and O-B (533.01 eV) [34,38,39]. The fine spectrum of B1s has four peaks, which are B-C (191.23 eV), B=O (192.3 eV), B2O3 (193.23 eV), and B-O (194.15 eV), respectively [35,40,41]. The above results indicate that the GQDs prepared by p-phenylenediamine, and boric acid contain the element B, indicating that B can be well doped into GQDs after the reaction. The peak in the XPS spectrum of the o-GQDs corresponds to their FT-IR diagram. The full XPS spectrum of the c-GQDs is shown in Appendix A
Appendix A. It can be clearly seen that it has three elements, namely, C, N and O, with the C1s peak at 285 eV, N1s peak at 399 eV, and O1s peak at 531 eV. Appendix AAppendix A is the fine spectrum of C1s, N1s, and O1s of the fitted c-GQDs. Their C1s fine spectrum has four peaks, their N1s fine spectrum has three peaks, and their O1s fine spectrum has four peaks. The atomic ratios of the c-GQDs and o-GQDs are shown in Appendix A
Appendix A. The C, N, O, and B contents of the o-GQDs are 3.3 %, 3.17 %, 4.24 %, and 3.63 %, while the C, N, and O contents of the c-GQDs are 3.5 %, 3.58 %, and 3.63 %.

### 2.3. Characterization of Optical Properties of GQDs

Figure 5a shows the ultraviolet absorption spectra and PL spectra of the two GQDs. The maximum absorption peaks of the c-GQDs and o-GQDs are 243 nm and 304 nm, respectively, which may be caused by the π–π* transition of the conjugated carbon–carbon double bond. As shown in the illustration in Figure 5b,c, the c-GQDs appear mauve under natural light, while o-GQDs appear brown under natural light. Under ultraviolet light, c-GQDs are cyan and o-GQDs are orange, indicating that after hydrothermal reaction, GQDs produce fluorescence. The best excitation wavelength of the c-GQDs is 360 nm and the best emission wavelength is 490 nm, while the best excitation wavelength of the o-GQDs is 480 nm and the best emission wavelength is 630 nm. As shown in Figure 5b,c, the fluorescence peak intensity of the c-GQDs and o-GQDs decreases or increases with the increase in the excitation wavelength, and the position of their peaks changes slightly, which may be due to the highly ordered graphite structure of the c-GQDs and o-GQDs; thus, the PL spectrum does not have excitation dependence.

In order to demonstrate the performance stability of the prepared GQDs, the pH stability, dispersion stability, and time stability of the two GQDs were also tested. Appendix A
Appendix A shows the test results of the pH stability of these two GQDs, and their initial pH is 7. It can be seen from Appendix A
Appendix A that the fluorescence intensity of the c-GQDs solution decreases slightly after adding different concentrations of alkali solution, indicating that the c-GQDs are alkali resistant. However, after adding different concentrations of acid, the fluorescence intensity decreases slightly at pH 1–2 and significantly at pH 3–6, especially plummeting at pH 5, indicating that the prepared c-GQDs have defects. It can be seen from Appendix A
Appendix A that after the o-GQDs solution is added to the alkali solution, the fluorescence intensity decreases increasingly more obviously with the increase in pH, and plummets when the pH is 14, indicating that the o-GQDs have alkali resistance, but not strong alkali resistance. When acid is added to o-GQDs, there is little difference in the fluorescence intensity at different pH levels. Compared with the initial pH of the o-GQDs, the fluorescence intensity decreases, but not sharply, indicating that the o-GQDs also have certain acid resistance. The performance of the o-GQDs is somewhat better than that of the c-GQDs, which may be the result of boron doping, which improves the performance of the o-GQDs and gives it better acid and alkali resistance.

Appendix AAppendix A shows the results of thermal stability test of the two GQDs. First, the fluorescence intensity of the 5 mL original GQDs solution was tested; then, the GQDs were dried, dispersed in the same amount of deionized water, and the fluorescence intensity was tested. Appendix A shows that the fluorescence intensity before and after for the o-GQDs and c-GQDs does not change significantly, indicating that they have good thermal stability. Appendix A
Appendix A shows the temporal stability of the two GQDs. The fluorescence intensities of the c-GQDs and o-GQDs were tested for seven consecutive days after they had been in place for three months and compared with their fluorescence intensities tested three months earlier. As can be seen from Appendix A, the stability of these two GQDs is very good, and the fluorescence intensity does not change much after remaining in place for three months.

### 2.4. GQDs for Ion Detection

In order to test the selective recognition of metal ions by the GQDs, a variety of metal ions were selected in this experiment; a 10 mM metal ion solution was prepared and then added into the c-GQDs and o-GQDs solution, the fluorescence intensity was tested, and fluorescence quenching was observed. As it can be seen from Figure 6a, after the c-GQDS solution is added to the metal ion solution, the fluorescence intensity does not decrease; this can also be seen from Figure 6b, i.e., that it did not reduce, and from Figure 6e in the fluorescence diagram, the c-GQDs solution, after joining all kinds of metal ions, shows almost no fluorescence quenching, which suggests that the c-GQDs does not apply to these metal ions detection. It can be seen from Figure 6a that the fluorescence intensity of the o-GQDs solution decreases after the addition of metal ion solution, especially Al^3+^. This conclusion can also be drawn from Figure 6c. Compared with other metal ions, Al^3+^ has the greatest influence on its fluorescence intensity. It can be seen from the fluorescence diagram in Figure 6e that compared with other metal ions, Al^3+^ has the most obvious fluorescence-quenching effect on the o-GQDs, while other ions have no obvious fluorescence-quenching effect on o-GQDs. Therefore, Al^3+^ was selected for ion detection test in this experiment.

The Zeta potentials of two GQDs at different pH values were tested in this study. They all had an initial pH of 7. As shown in Appendix A
Appendix A, when the acid was added, the Zeta potential increased with the H^+^ ions, and when the base was added, the Zeta potential decreased with the OH^–^ ions. Subsequently, the functional groups in GQDs neutralize with the addition of H^+^ ions or OH^–^ ions, flattening the Zeta potential. When pH is neutral, the Zeta potential of the c-GQDs is 2, indicating that the c-GQDs are neutral, and the Zeta potential of o-GQDs is 10, indicating that o-GQDs have a positive charge. Since boron is doped into the o-GQDs, the boron atoms have a positive charge, causing the o-GQDs to have a positive charge. The positively charged boron breaks the electrical neutrality of the sp^2^ carbon, creating a charge site that promotes the interaction between o-GQDs and Al^3+^. After the addition of Al^3+^, Al^3+^ adsorbs to the surface of the o-GQDs, inducing most of the excited electrons to transfer to the lowest energy level orbital (LUMO) of Al^3+^, which promotes the electron/hole recombination process, causes the electron transition to be hindered, and finally leads to the quenching of the fluorescence signal of o-GQDs [42,43].

The fluorescence intensity of o-GQDs and their linear relationship with Al^3+^ concentration was tested for the mixed solution with Al^3+^ concentration increasing from 0 μM to 1.5 μM. As shown in Figure 7a, when the concentration of Al^3+^ ions increases from 0 μM to 1.5 μM, the fluorescence intensity of the o-GQDs solution at 480 nm gradually decreases. In addition, it can be seen from Figure 7b that there is a good linear relationship between the Al^3+^ ion concentration and the o-GQDs PL fluorescence intensity. The linear regression Equation (1) is:y = 20.04114 − 3.79994x(1)
and the correlation coefficient R^2^ is 0.9937. Appendix A
Appendix A shows the linear relationship between the concentration of Al^3+^ from 0 to 1.5 μM and the PL fluorescence intensity of the o-GQDs. As can be seen from Appendix A, there is a good linear relationship between the fluorescence intensity and Al^3+^ concentration from 0 μM to 1.5 μM. Appendix A
Appendix A shows the linear relationship between the concentration of Al^3+^ from 0 to 12 μM and the PL fluorescence intensity of the o-GQDs. As can be seen from Appendix A, when the concentration of Al^3+^ increases, the linear relationship between the PL fluorescence intensity and Al^3+^ concentration becomes worse, indicating that with the increase in the Al^3+^ concentration, the o-GQDs could not detect Al^3+^ well. Then, the reversible cycle of the o-GQDs in the presence of Al^3+^ and EDTA was also tested with respect to fluorescence intensity. As shown in Appendix A
Appendix A, EDTA can bleach the signal emission band of Al^3+^, and the fluorescence intensity can be completely restored by adding EDTA to the GQDs solution mixed with ionic solution. Reversibility can be observed from several cycles in Appendix A.

## 3. Materials and Methods

### 3.1. Experimental Material

P-phenylenediamine (p-PD), citric acid (CA), and boric acid (BA) were purchased from Shanghai Sinopharm Chemical Reagent Co, LTD. FeCl_3_, CoCl_2_, KNO_3_, CuCl_2_, Cd(NO_3_)_2_, LiCl, MnCl_2_, NiCl_2_, NaCl, ZnCl_2_, Al(NO_3_)_3_, and MgCl_2_ were purchased from national pharmaceutical institutions.

### 3.2. Experimental Facilities

Experimental materials were weighed by Electronic Balance (Model is FA2004B, purchased from Shanghai Youke, Shanghai, China). GQDs solution was filtered through disposable needle filters (25–0.22 μm). The deionized water used in the experiment comes from a Pure water/Ultrapure water manufacturing instrument (Model is Smart-Q15, purchased from Sichuan Youpu Chaopure Technology Co., Ltd., Chengdu, China.). Solution and solvent could be fully dissolved by CNC ultrasonic cleaning device (KQ-300DA, purchased from Kunshan Ultrasonic Instrument Co., Ltd., Suzhou, China). The reaction took place in an Electric Blower drying oven (Model is DHG-9036A, purchased from Shanghai Yi Heng Scientific Instrument Co., Ltd., Shanghai, China). The fluorescent color of GQDs was observed under High intensity UV lamp (FC-100/FA, Spectroline Company, Melville, NY, USA). Raman spectra were tested by Confocal Microraman Spectrometer (Horiba Xplora Plus, Paris, France). AFM were tested by Atomic Force microscope (MFP-3D Infinity, Oxford, UK). Zeta potential was tested by Laser particle size analyzer (Nicomp 380 Z3000 SOP, Entegris, Billerica, MA, USA). The infrared spectrum was tested by Fourier Transform Infrared spectrometer (Bruker TENSOR37, Bremen, Germany). XRD was texted by Multi-function X-ray Diffractometer (D/MAX2500V+/PC, Japan Electronics Co., Ltd., Tokyo, Japan). The ion detection process required the use of a Micropipette gun (20–200 μL, 100–1000 μL, Eppendorf AG, Hamburg, Germany). The fluorescence intensity was tested by the Fluorescence spectrophotometer (RF-6000, Shimadzu Corporation, Shimadzu, Japan). TEM was tested by Transmission electron microscope (JEM-2010F, Japan Electronics Co., LTD., Tokyo, Japan). UV–Visible absorption spectra were measured by UV–Visible near-infrared spectrophotometer (LAMBDA750, PerkinElmer, Waltham, MA, USA).

### 3.3. Synthesis of GQDs

Take 0.1 g of p-PD and 0.05 g of boric acid or 0.05 g of CA citric acid, mix them together, add 10 mL of deionized water, apply ultrasonic shocking, and ensure that it is fully dissolved. Then, transpose this mixture to a 10 mL PTFE stainless steel reactor for hydrothermal reaction, and ensure that the hydrothermal reaction temperature is 180 °C and that the reaction time is 12 h. After the reaction is completed, the reactor should be cooled to room temperature and removed, and the reaction liquid must be filtered by a microporous membrane at 220 nm to obtain GQDs solution. Name them o-GQDs (BA was used as acid regulator) and c-GQDs (CA was used as acid regulator), respectively.

### 3.4. Metal Ion Detection

In this paper, some metal ions (Al^3+^, Ni^2+^, Mg^2+^, Zn^2+^, Cd^2+^, Mn^2+^, K^+^, Na^+^, and Li^+^) were selected to detect the effect of metal ions on the fluorescence intensity of c-GQDs and o-GQDs. The concentration of the metal ion solution was uniformly set at 10 mM; then, a certain amount of metal ion solution was mixed with the c-GQDs and o-GQDs, and the fluorescence-quenching degree of the different concentrations of metal ions on the quantum dot solution was tested.

## 4. Conclusions

In this study, orange, fluorescent o-GQDs were prepared by the bottom-up solvothermal method with p-phenylenediamine as a precursor and boric acid as a regulator. The optical properties, morphology, and structure of the prepared o-GQDs were characterized. The results show that the prepared o-GQDs have excellent properties and have the characteristics of graphene quantum dots. At the same time, o-GQDs were detected by different ions. It was found that fluorescence-quenching activity of the o-GQDs was obvious under the action of Al^3+^, indicating that o-GQDs can detect Al^3+^. The application of GQDs in the field of chemical sensing will become increasingly extensive, and we believe that graphene quantum dots will be widely used and understood by increasingly more people.

## Figures and Tables

**Figure 1 molecules-27-06771-f001:**
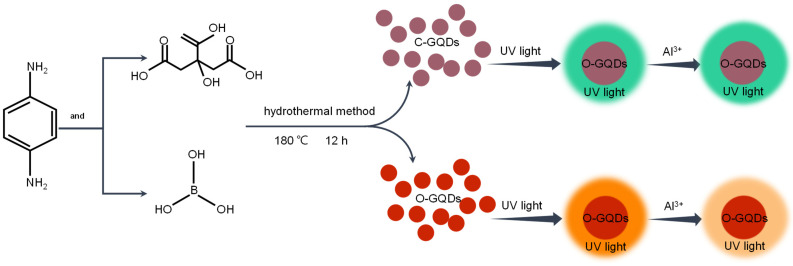
Preparation flow chart of c-GQDs and o-GQDs.

**Figure 2 molecules-27-06771-f002:**
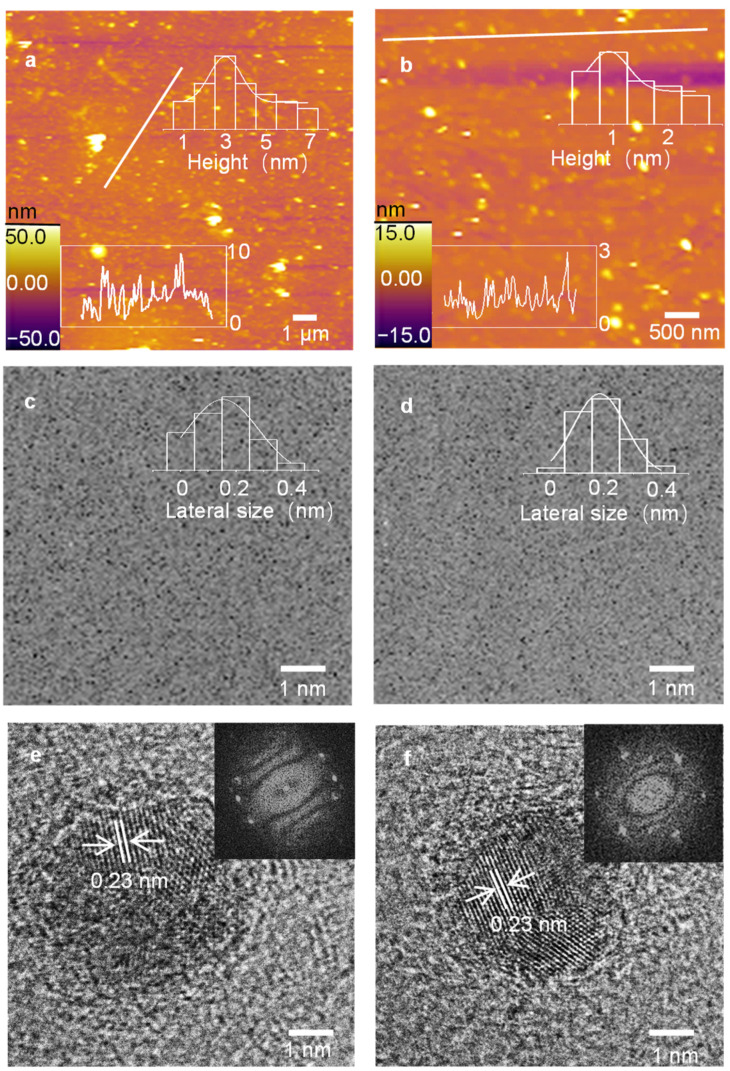
(**a**,**b**) AFM diagram and height distribution of c-GQDs and o-GQDs. (**c**,**d**) TEM image and relative size distribution of c-GQDs and o-GQDs. (**e**,**f**) High resolution TEM images of c-GQDS and o-GQDs (inset: Fast Fourier transform mode).

**Figure 3 molecules-27-06771-f003:**
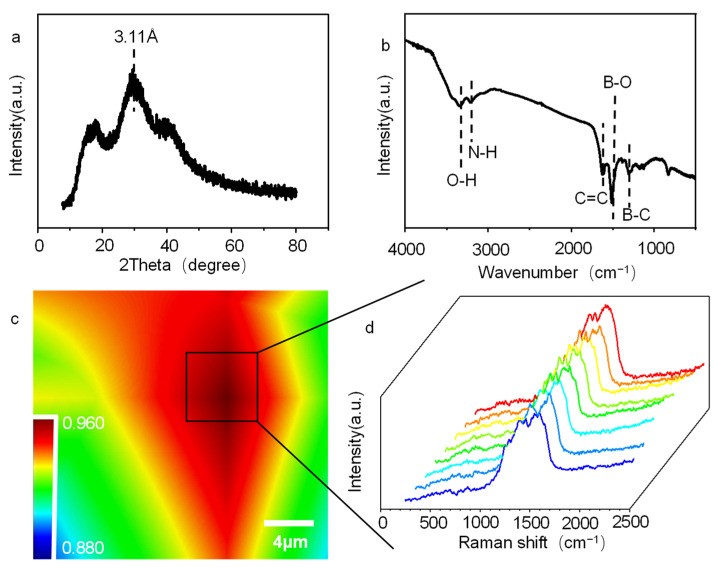
(**a**) XRD patterns of o-GQDs, (**b**) FT-IR patterns of o-GQDs, (**c**) High resolution microscope Raman image of c-GQDS (color indicates the ratio of peaks D to G), and (**d**) Typical Raman spectra in the box region of Figure 3d.

**Figure 4 molecules-27-06771-f004:**
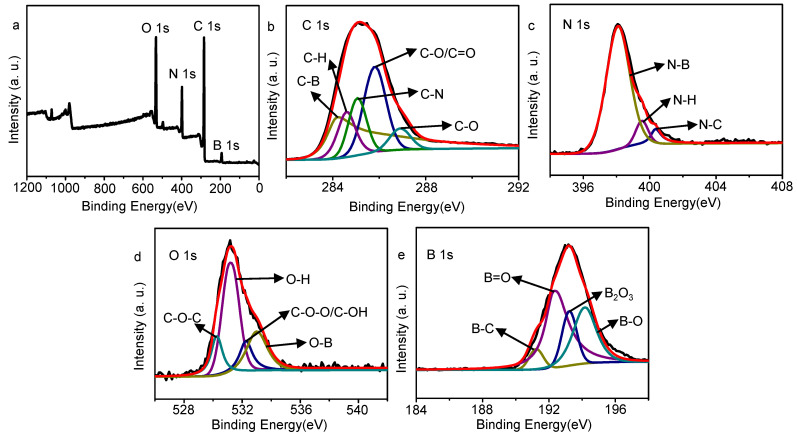
(**a**) Full X-ray photoelectron spectrum of o-GQDs, (**b**) C1S spectrum measured by X-ray photoelectron spectrum of o-GQDs, (**c**) N1s spectrum of o-GQDs, (**d**) O1s spectrum of o-GQDs, and (**e**) B1s spectrum of o-GQDs.

**Figure 5 molecules-27-06771-f005:**
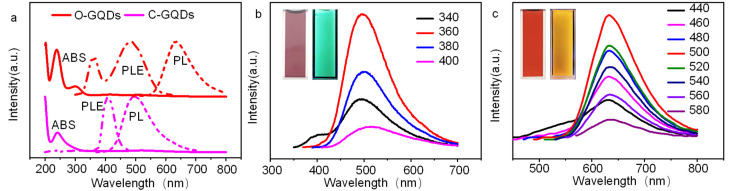
(**a**) UV–Vis absorption spectra and PL spectra of o-GQDs and c-GQDs, (**b**) FL spectra of c-GQDs at different excitation wavelengths, and (**c**) FL spectra of o-GQDs at different excitation wavelengths.

**Figure 6 molecules-27-06771-f006:**
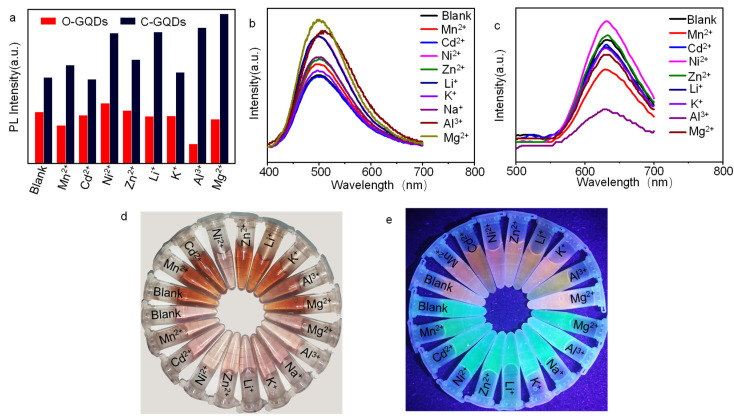
(**a**) The difference in the relative PL strengths of c-GQDs and o-GQDs between blank and solutions containing different metal ions, (**b**) Spectrograms of c-GQDs containing different metal ion solutions, (**c**) spectrograms of o-GQDs containing different metal ion solutions, and (**d**,**e**) photographs of c-GQDs and o-GQDs containing different metal ion solutions under natural light and ultraviolet light (the c-GQDs solution is lavender under natural light and cyan under UV light, and the o-GQDs solution is brown under natural light and orange under UV light).

**Figure 7 molecules-27-06771-f007:**
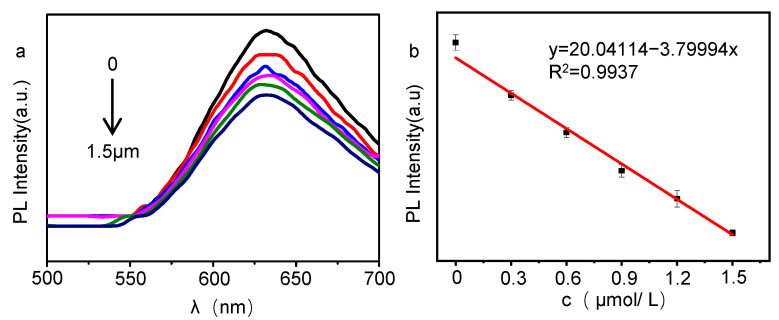
(**a**,**b**) PL intensity changes of o-GQDs at different concentrations of Al^3+^.

## Data Availability

The data can be made available upon reasonable request.

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
