# Peer review of "Fabrication of Orange Fluorescent Boron-Doped Graphene Quantum Dots for Al3+ Ion Detection"

_molecules, 2022, doi:10.3390/molecules27196771_

Round 1

Author Response

Thanks a lot for your valuable review. We have responded to the questions one by one, please see the attachment for details.

Author Response

(The authors gave the same response as above.)

Round 2

Reviewer 1 Report

The authors positively answer and the modifications of the manuscript improve the quality of their work. Additionally, the improvement of the supplementary information, especially the comparison between the two quantum dots increase the performance of the paper.

Reviewer 2 Report

In the revised manuscript the issues raised have been sufficiently addressed and overall the quality of the work has been improved. I suggest the acceptance of the manuscript.